# Application of Machine Learning for Differentiating Bone Malignancy on Imaging: A Systematic Review

**DOI:** 10.3390/cancers15061837

**Published:** 2023-03-18

**Authors:** Wilson Ong, Lei Zhu, Yi Liang Tan, Ee Chin Teo, Jiong Hao Tan, Naresh Kumar, Balamurugan A. Vellayappan, Beng Chin Ooi, Swee Tian Quek, Andrew Makmur, James Thomas Patrick Decourcy Hallinan

**Affiliations:** 1Department of Diagnostic Imaging, National University Hospital, 5 Lower Kent Ridge Rd, Singapore 119074, Singapore; 2Department of Computer Science, School of Computing, National University of Singapore, 13 Computing Drive, Singapore 117417, Singapore; 3University Spine Centre, Department of Orthopaedic Surgery, National University Health System, 1E, Lower Kent Ridge Road, Singapore 119228, Singapore; 4Department of Radiation Oncology, National University Cancer Institute Singapore, National University Hospital, 5 Lower Kent Ridge Road, Singapore 119074, Singapore; 5Department of Diagnostic Radiology, Yong Loo Lin School of Medicine, National University of Singapore, 10 Medical Drive, Singapore 117597, Singapore

**Keywords:** deep learning, machine learning, artificial intelligence, bone malignancy, imaging

## Abstract

**Simple Summary:**

Distinguishing between benign vs. malignant bone lesions is often difficult on imaging. Many bone lesions are infrequent or rarely seen, and often only specialist radiologists have sufficient expertise to provide an accurate diagnosis. In addition, some benign bone tumours may exhibit potentially aggressive features that mimic malignant bone tumours, making the diagnosis even more difficult. The rapid development of artificial intelligence (AI) techniques has led to remarkable progress in image-recognition tasks, including the classification and characterization of various tumours. This study will review the most recent evidence for AI techniques in differentiating bone lesions on various imaging modalities using a systematic approach. Potential clinical applications of AI techniques for clinical diagnosis and management of bone lesions will also be discussed.

**Abstract:**

An accurate diagnosis of bone tumours on imaging is crucial for appropriate and successful treatment. The advent of Artificial intelligence (AI) and machine learning methods to characterize and assess bone tumours on various imaging modalities may assist in the diagnostic workflow. The purpose of this review article is to summarise the most recent evidence for AI techniques using imaging for differentiating benign from malignant lesions, the characterization of various malignant bone lesions, and their potential clinical application. A systematic search through electronic databases (PubMed, MEDLINE, Web of Science, and clinicaltrials.gov) was conducted according to the Preferred Reporting Items for Systematic Reviews and Meta-Analyses (PRISMA) guidelines. A total of 34 articles were retrieved from the databases and the key findings were compiled and summarised. A total of 34 articles reported the use of AI techniques to distinguish between benign vs. malignant bone lesions, of which 12 (35.3%) focused on radiographs, 12 (35.3%) on MRI, 5 (14.7%) on CT and 5 (14.7%) on PET/CT. The overall reported accuracy, sensitivity, and specificity of AI in distinguishing between benign vs. malignant bone lesions ranges from 0.44–0.99, 0.63–1.00, and 0.73–0.96, respectively, with AUCs of 0.73–0.96. In conclusion, the use of AI to discriminate bone lesions on imaging has achieved a relatively good performance in various imaging modalities, with high sensitivity, specificity, and accuracy for distinguishing between benign vs. malignant lesions in several cohort studies. However, further research is necessary to test the clinical performance of these algorithms before they can be facilitated and integrated into routine clinical practice.

## 1. Introduction

Differentiating between benign vs. malignant bone tumours is critical for clinical decision-making and treatment planning [1,2]. Routine imaging investigations for evaluating bone lesions include radiographs (X-ray), computed tomography (CT), positron-emission tomography combined with computed tomography (PET/CT), and magnetic resonance imaging (MRI) [3,4]. Conventional radiography remains a key initial imaging modality and is still an optimal technique for evaluating primary bone tumours [5,6]. It is relatively inexpensive and allows for a clear visual assessment of lesion location, margins, internal lesion matrix, and any associated periosteal reaction [7]. Along with the patient’s age, these radiographic details are often sufficient to provide a reasonable list of differential diagnoses [8]. However, radiographs are often limited by superimpositions, incomplete visualizations of bone cortex destruction, and inadequate assessments of adjacent soft tissue involvement [9,10,11]. Furthermore, diagnosis of bone tumours using imaging can be complicated by other factors such as the presence of pathological fractures, which may result in benign bone lesions having potentially aggressive features that mimic malignant bone tumours [12,13].

Multi-detector CT can be a useful adjunct to radiographs as it allows precise delineation of complex anatomical locations including articulations [3] and irregular bones, such as the sacrum or vertebral bodies [14]. These regions are often difficult to visualize in 2D (2-Dimensional) planes and CT can provide additional 3D (3-Dimensional) reconstructions [15], which are useful for surgical planning. The evaluation of minor bone changes, such as tumour mineralization, bone cortex changes, and periosteal reaction, are also better depicted on CT scans [16]. In addition, CT provides simultaneous evaluation for both bone and soft tissue lesions in cases of suspected malignancy (systemic staging), which reduces the burden of imaging for patients [17,18]. However, CT is deficient in evaluating the soft tissue extent of bone lesions, as well as the degree of medullary involvement [19,20].

MRI is a highly sensitive modality for the delineation of bony abnormalities and provides the ability to assess bone marrow involvement, the extent of soft tissue invasion, and the internal content of the bone lesions [21]. Occasionally, the excellent tissue or lesion characterization provided by MRI can yield sufficient information to clinch an accurate diagnosis, even without histological correlation [20,22]. With the advent of functional MRI sequences, which include perfusion or dynamic contrast-enhanced MR imaging, Diffusion Weighted Imaging (DWI), and MRI Spectroscopy, there is even more potential for the accurate differentiation between benign vs. malignant bone lesions [23,24,25,26,27].

For radiographically indeterminate bone lesions, ^18^F-Flurodeoxyglucose Positron Emission Tomography combined with Computed Tomography (18F-FDG PET/CT) has been reported to provide an improved differentiation between benign vs. malignant bone lesions in comparison to CT or MRI alone [2,28,29]. FDG avidity of the bone lesion helps to predict aggressiveness, with a malignant tumour showing increased avidity relative to a benign bone lesion of the same histological subtype [30,31]. However, FDG uptake within a bone lesion does not usually help to determine the morphologic features and specific subtype of the bone tumour. Moreover, FDG uptake in bone lesions can lead to false positives for malignancy, with superimposed trauma or fractures, background bony hyperplasia, and underlying metabolic bone disease is also reported to increase the FDG uptake within bone lesions, thus, confounding the assessment [32,33,34].

Even when combining various imaging modalities, radiologists are often not accurate or specific in classifying bone lesions [35]. Considering the potential limitations of current advanced imaging and the wide spectrum of bone tumours encountered in clinical practice, there is a clear utility for emerging technologies to aid clinicians in the detection and characterization of benign from malignant bone tumours.

Emerging Artificial Intelligence (AI) tools continue to demonstrate remarkable progress in medical imaging applications, especially in the field of oncology [36]. These applications include cancer screening and diagnosis [37,38,39,40], diagnosis and classification [41,42,43,44], predicting prognosis and treatment response [45,46,47,48,49], automated segmentation [50,51,52,53,54], and radiology-pathology correlation (radiogenomics) [55,56,57,58]. In particular, within the field of diagnosis and classification, the ability of AI models to classify benign vs. malignant tumours has been shown to achieve high accuracy, sensitivity, and specificity in various organs, such as in the case of breast [59,60,61], prostate [62,63], lung [38,64,65,66], and brain lesions [67,68]. This review article aims to provide an overview of the current evidence on the effectiveness of machine learning in differentiating bone lesions on various imaging modalities.

## 2. Materials and Methods

### 2.1. The Literature Search Strategy

A systematic search of the major electronic databases (PubMed, MEDLINE, Web of Science, and clinicaltrials.gov, accessed on 31 December 2022) was conducted in concordance with the Preferred Reporting Items for Systematic Reviews and Meta-Analyses (PRISMA) guidelines [69,70] using keywords and Medical Subject Headings (MeSH), or both, for the following key terms: (“bone” OR “vertebral” OR “spinal”) AND (“tumour” OR “lesion” OR “malignancy” AND (“radiomics”, OR “machine learning”, OR “artificial intelligence”, OR “deep learning”). Two authors (W.O. and Y.L.T.) performed an independent review of the collected references and selected the appropriate studies for detailed full-text screening.

### 2.2. The Study Screening and Selection Criteria

No limitations were specified for the reference and literature search. The main inclusion criteria were scientific studies harnessing radiomic techniques, Artificial Intelligence (AI), or deep learning to distinguish bony lesions. Inclusion criteria also involved the following: (a) imaging analysis involving radiographs (X-rays), nuclear medicine (radioisotope) imaging, Computed Tomography (CT, all types including Positron Emission Tomography combined with Computed Tomography (PET/CT)), and Magnetic Resonance Imaging (MRI) scans; (b) studies that addressed the ability to differentiate between benign vs. malignant bone (primary malignant or metastatic) lesions (c) studies involving human subjects only; and (d) publications primarily in the English Language. Articles that were excluded from the further analysis included case reports, editorial correspondence (for example, opinion pieces, letters, and commentaries), and review articles. Duplicate publications, those focusing on non-imaging (for example, genomic) radiomic AI techniques or articles on the applications of AI technology not related to distinguishing bony lesions (for example, segmentation and detection, among others) were also excluded from this analysis.

### 2.3. Data Extraction and Reporting

All selected research articles were retrieved and compiled into a spreadsheet using Microsoft Excel (Microsoft Corporation, Washington, DC, USA). Information gathered from the individual research articles included:Research article details: Complete authorship, date of journal or publication, Journal name;Main clinical use: Differentiating benign vs. malignant bone lesions, characterization and classification of various bone tumours;Patient population: Patients with known bone lesions, who have undergone various imaging investigations (X-ray, CT, PET/CT or MRI) and have subsequently undergone histopathological confirmation;Research study details: The type of study, patient or imaging modality sample sizes (for example, internal or external data sets), imaging modalities used (CT, MRI, bone scans or PET/CT), treatment or management information and outcome/prognostic measures;Machine Learning techniques used: Radiomics and convolutional neural networks, among others.

## 3. Results

### 3.1. Search Results

The primary search through the relevant electronic medical databases (please see Figure 1) identified a total of 54 relevant research articles, which were initially screened using the previously detailed criteria. This initial screening led to the exclusion of eight publications, and the remaining 46 articles then underwent a detailed full-text analysis to determine their inclusion or exclusion. Following the detailed full-text analysis, a further 25 publications were excluded from further analysis as they were either focused on cancer or lesion sites other than bone or related to other Artificial intelligence (AI) applications of bone lesions and were not related to differentiating or characterizing them (for example, detection, segmentation or prognosis). An additional 13 articles were included after the study team manually reviewed the bibliography of the selected research manuscripts. Overall, this screening yielded a total of 34 publications (please see Table 1) for detailed analysis. Key findings from each article were then compiled, categorized and summarized for this review. Most studies lacked the detailed data required to create 2 × 2 contingency tables, hence a formal meta-analysis could not be performed.

Our search found that 12/34 (35.3%) studies were X-ray-based, 5/33 (14.7%) were CT (Computed Tomography) based, 5/33 (14.7%) were related to nuclear studies (for example, Positron Emission Tomography with CT (PET-CT)), and 12/33 (35.3%) were MRI (Magnetic Resonance Imaging) based. All studies were retrospective in nature using radiological images fed into various AI systems, with the majority of patients having either a confirmed pathological diagnosis or an agreed consensus of the clinical diagnosis.

The overall reported accuracy, sensitivity, and specificity of AI in distinguishing between benign vs. malignant bone lesions (Table 2) ranges from 0.44–0.99, 0.63–1.00, and 0.73–0.96, respectively. The studies on X-rays reported an accuracy of 0.44–0.99, a sensitivity of 0.75–1.00, and a specificity of 0.78–0.91 with an AUC of 0.79–0.95. A note is made of a high sensitivity of 1.00 achieved in a study by Consalvo et al. [97] using radiographs to differentiate between acute osteomyelitis vs. malignant Ewing sarcoma. This high sensitivity may be related to data misbalance due to the small sample size and the use of relatively “normal” radiographs as a healthy control group. For CT studies, the reported accuracy, sensitivity, and specificity range from 0.74–0.92, 0.80, and 0.96, respectively, with an AUC of 0.78–0.96. PET/CT studies reported an accuracy of 0.74–0.88, a sensitivity of 0.84–0.90, and a specificity of 0.74–0.85 with an AUC of 0.76–0.95. Last but not least, the included MRI studies reported overall accuracy ranging from 0.74–0.98, sensitivity from 0.78–0.88, and a specificity of 0.61–0.79 with an AUC of 0.73–0.94.

Of note, studies with a two-label classification (benign vs. malignant) achieved relatively higher performance when compared to studies with three or more label classifications. For example, Pan D. et al. [84] showed that the ability of their AI model for binary classification (non-malignant vs. malignant) on radiographs achieved a higher AUC of 0.97 and an accuracy of 0.95 when compared to tertiary classification (benign vs. intermediate vs. malignant), with an AUC of 0.94 and an accuracy of 0.83. Similarly, the study performed by Chianca et al. [95] showed that their machine learning model achieved a higher accuracy of 0.86, vs. 0.68, when classifying vertebral lesions into a dichotomous (benign or malignant) compared to trichotomous classification (benign, primary malignant or metastases), respectively.

### 3.2. Machine Learning Techniques

AI refers to the computational ability of a machine to execute tasks to a level comparable to those performed by humans. This is achieved by utilizing unique data inputs to generate outputs that have a high-added value [104]. Recent advances in the field of medical imaging along with the availability of large volumes of imaging and report data have fueled worldwide interest in the use of AI techniques for medical imaging [105]. The initial rationale for developing and deploying AI tools was to assist clinicians (mainly radiologists) in the detection and characterization of lesions, which would have the benefit of increasing efficiency and detection accuracy, and reducing diagnostic errors [106]. With recent advances in computing and AI, many other applications, including the characterization of lesions, automated segmentation, and decision support planning (for example, predicting phenotypes and outcomes) have been studied [107].

Machine learning represents a subset of AI where models are trained for the prediction of outcomes using datasets with known ground truth, from which the machine “learns”. The developed AI model can then apply its new knowledge to predict outcomes in previously unseen datasets [108]. Radiomics is a new division of machine learning which involves converting the information stored within medical images into measurable and quantifiable data [109]. The information obtained can then be used to aid radiologists and clinicians in the assessment of benign and malignant lesions by analyzing and utilizing additional details beyond that of visual interpretation (inferable by the human eye) [110,111]. When radiomics is combined with additional clinical and qualitative imaging data or both, it has the potential to guide and optimize clinical decision-making, including improved lesion detection, classification, prognostication, and enhanced treatment response assessments [112]. In general, the typical workflow for developing a radiomics model can be divided into the following steps (Figure 2): image acquisition, data selection (image input), segmentation, image feature extraction within the curated regions of interest (ROIs), exploratory analysis with feature selection, and modelling [113]. The models should then be validated using test sets (preferably using both internal and external data) to evaluate their performance [114].

The two most common machine learning methods (Figure 3) are radiomics-based imaging feature analysis and convolutional neural networks (CNN). Feature-based radiomics techniques involve the extraction of various handcrafted features, which can then be included in a training set for AI-based imaging classification [115]. On the other hand, CNNs utilize deep learning to extract useful imaging features by learning their patterns, classifying data directly from input images [116], and transforming them into useful outputs [117,118]. This results in the ability to detect and process distinct diagnostic patterns and imaging features beyond that of human readers, which are then used for various applications including lesion detection, characterization, and providing prognostic information [119].

## 4. Discussion

### 4.1. Machine Learning on Conventional Radiographs

Correctly classifying bone tumours on conventional radiographs is vital for clinical decision-making and guiding subsequent management [120]. However, this is often difficult, especially in places where there is a shortage of subspecialty radiology expertise. Moreover, many bone lesions are uncommon entities, and often only a few specialist radiologists have sufficient experience to diagnose them accurately [10,79]. In clinical practice, most radiologists rely on image pattern recognition to distinguish between benign and malignant lesions on radiographs, which is subject to bias and can sometimes lead to an erroneous interpretation [121,122]. Some of these common radiological features include location, cortical destruction, periostitis, lesion orientation or alignment, and the zone of transition between the lesion and the surrounding bone [123,124,125,126]. However, some benign bone lesions may demonstrate one or more aggressive features which may confound the distinction [127,128].

Machine learning techniques can identify the more important radiographic features for distinguishing between benign vs. malignant bone lesions. Pan D et al. [84] performed a study using random forest (RF) models and identified that the most important imaging features to distinguish between benign vs. malignant bone lesions, in descending order of importance, are: margins, cortical bone involvement, the pattern of bone destruction, and the internal high-density matrix, with an accuracy of up to 94.7% and an area under the curve (AUC) of 0.97. A deep learning algorithm developed by He et al. [79] achieved high performance using a multi-institutional dataset comprised of up to 2899 images with pathologically proven bone tumours (benign 52.5%, vs. intermediate 21.9%, vs. malignant 25.6%). The developed model demonstrated an AUC reaching up to 0.92 and an accuracy of 72.1% for trichotomous classification, as per the World Health Organization (WHO) classification of bone tumours [129]). The model outperformed two junior radiologists (accuracies 63.4%–67.9%, *p* < 0.05) and was similar in accuracy compared to two subspecialist radiologists (accuracy 69.3%–73.4%, *p* > 0.1). Similarly, a deep learning model created by Liu et al. [83] achieved diagnostic performance comparable to that of senior radiologists for benign, intermediate, and malignant bone lesions. The developed fusion model combined clinical and imaging features and achieved an AUC of 0.87 vs. 0.82 for the radiologists, which did not reach statistical significance (*p* = 0.86).

Malignant bone lesions are often difficult to differentiate from other aggressive disease processes, including inflammation and infection. Ewing’s sarcoma, an aggressive malignant tumour occurring in children, is a typical example [130]. Differentiating this entity from acute osteomyelitis is often difficult, even by trained musculoskeletal radiologists, due to its similar clinical and radiological features [131,132,133]. Consalvo et al. [97] developed an artificial intelligence algorithm which was able to leverage radiographic features to distinguish between Ewing sarcoma and acute osteomyelitis, achieving an accuracy of up to 94.4% on their validation set and 90.6% on a held-out test set. Although this study is limited by a small sample size, requiring the use of cross-validation and loss weighting to achieve statistical significance, it demonstrates the potential feasibility of AI techniques for differentiating infective bony lesions from malignant bone lesions on routine radiographs.

### 4.2. Machine Learning on Computed Tomography (CT) Imaging

Incidentally detected dense (sclerotic) lesions are a common occurrence on CT examinations in clinical practice [134]. The ability to distinguish a benign sclerotic lesion, such as an enostosis (bone island), vs. a malignant lesion, such as an osteoblastic metastasis, is crucial as it affects the treatment strategy and patient prognosis [135]. For this task, radiomics-based random forest models were created by Hong et al. [85] and achieved an AUC of up to 0.96 and an accuracy of up to 86.0% in the test sets, which was comparable to two experienced radiologists (AUCs of 0.95–0.96) and even higher compared to a radiologist in training (AUC 0.88, *p* = 0.03). Along with the extraction of 1218 radiomics features, the authors showed that a model utilizing attenuation and shape-related features achieved the highest AUC, which had been postulated in several prior studies [136,137,138,139,140]. In another study, Sun et al. [74] developed a CT-derived nomogram using radiomics to characterize benign vs. malignant bone tumours. This utilized radiomics features from the texture analysis, including a ground-glass matrix, peripheral lesion sclerosis, residual bony ridge, and the presence of a soft tissue mass [141,142], and demonstrated an AUC of up to 0.92. In addition, the team also showed that by including clinical features in the nomogram, the model achieved higher net clinical benefits for decision-making when compared to radiomics alone with an NRI (Net reclassification index) of 0.24 (95% CI 0.07–0.41) and an IDI (Integrated discrimination index) of 0.16 (95% CI 0.11 to 0.22), albeit not reaching statistical significance.

Besides differentiating between benign vs. malignant bony lesions, radiomics models can also be used to differentiate between a variety of tumour matrix types with high performance. A deep convolutional neural network (CNN) created by Y. Li et al. [81] was able to further classify benign and malignant bone lesions into cartilaginous or osteogenic tumours using a multi-channel enhancement strategy for image processing to improve accuracy, achieving a top-1 error rate of only 0.25. In a similar study that focused on radiographs rather than CT, Reicher et al. [80] created a recurrent CNN model which was able to learn and classify the bone tumour matrix with a high accuracy of 93% compared to the average radiologist’s accuracy of 70%. This shows the potential use of artificial intelligence to further discriminate various subtypes of bone tumours (for example, chondroid or osteoid tumours) which is important for biopsy and treatment planning.

### 4.3. Machine Learning on Magnetic Resonance Imaging (MRI)

MRI plays a key role in aiding clinicians in discriminating between benign vs. primary malignant or metastatic bone tumours [143]. The conventional pulse sequences [20,144], diffusion-weighted imaging (DWI) [145,146] with matched apparent diffusion coefficient (ADC) maps [147,148] as well as dynamic contrast sequences (DCE) [149,150] can predict potential malignancy with good reliability. However, some imaging features of benign vs. malignant bone lesions can overlap and this makes formulating a differential diagnosis challenging [151]. Machine learning techniques using radiomic features on MRI have been shown to have high performance for predicting benign vs. malignant bone lesions. Using pre-trained ResNet50 image classifiers, Georgeanu et al. [93] were capable of predicting the malignant potential of bone tumours in 93.7% of cases using T1-weighted sequences and 86.7% using T2-weighted sequences. A model developed by Chianca et al. [95] for classifying vertebral lesions into benign vs. malignant (primary malignant and metastatic lesions) demonstrated 94.0% accuracy in the internal test dataset and 86% accuracy in an external validation dataset. This showed no significant difference relative to an expert musculoskeletal radiologist with more than 5 years-experience who achieved 94.0% in the internal test cohort (*p* = 0.99). However, there was no consistent MRI protocol or sequences among the different studies, and a wide variety of software was used for image segmentation and feature extraction. In addition, there was inconsistency in the number and type of features explored and the type of deep learning models used to classify the final features. Overall, these factors may have reduced the reproducibility of the results. Future multicenter validation studies could be performed with more standardized protocols to assess the generalizability of the deep learning applications and facilitate their integration into routine clinical practice.

Specific to cartilaginous bone lesions, machine learning has been studied for the differentiation of various grades of chondrosarcoma. Conventional chondrosarcoma is usually divided into three categories based on pathology, where grade 1, also known as atypical cartilaginous tumours (ACTs), usually have an indolent biologic nature, whereas grades 2–3 (high-grade chondrosarcoma) are malignant bone tumours [152] with metastatic potential and a high recurrence rate following surgical resection [153]. Discrepancies in correct tumour grading are widespread even among experienced radiologists and pathologists, secondary to overlapping imaging and histological features, and it is for this reason that more accurate diagnostic aids are required [154,155]. Gitto et al. [91] used MRI-derived radiomics to characterize ACT vs. high-grade chondrosarcoma, based on conventional T1- and T2-weighted images, and achieved 85.7% and 75.0% accuracy (AUCs of 0.85 and 0.78) in the training and test groups, respectively, with no difference in diagnostic performance between the radiologist and machine learning classifier (*p* = 0.45). Further to that, Gitto et al. [90] utilized a similar radiomics-based MRI method to discriminate between ACT and Grade II chondrosarcoma, achieving 92% accuracy (AUC of 0.94) with no significant difference compared to an expert radiologist (*p* = 0.13). With the help of AI, the discrimination between various grades of bone tumours could be improved, although further external multicenter validation is necessary to assess the generalisability of the proposed models before they can be applied in a prospective clinical setting.

The use of intravenous contrast media (gadolinium-based) for the evaluation of bone tumours has been shown to add some specificity in tissue characterization [156,157], although the main advantages are for the accurate evaluation of tumour extent for biopsy and treatment planning and to assess for recurrence [158,159,160]. Interestingly, a study by Eweje et al. [72] demonstrated that a deep learning model was able to achieve a performance similar to that of expert radiologists for classifying bone lesions at various skeletal locations without using post-contrast T1-weighted sequences, which were made available to the radiologists. The model had an accuracy of 76% vs. 73% for radiologists (*p* = 0.7) for classifying benign (which includes intermediate as per WHO classification criteria) vs. malignant bone lesions. This preliminary study shows the potential utility of machine learning for the accurate diagnosis of bone tumours without requiring the administration of gadolinium-based MRI contrast media. This could be advantageous, especially in patients who have contraindications to gadolinium-based contrast (due to renal impairment or allergy) or in children with pain-related anxiety that is secondary to the placement of an intravenous cannula and the uncertain long-term implications of gadolinium deposition in children [161,162]. Further larger studies are required to show if machine learning using a combination of non-contrast and contrast-enhanced imaging has an advantage over non-contrast imaging alone.

### 4.4. Machine Learning on Positiron Emission Tomography with CT (PET/CT) Imaging

PET/CT imaging is a widely used modality to differentiate malignant vs. benign tumours in a host of organ systems [163,164,165]. The most common radiotracer used in PET/CT imaging is 2-deoxy-2-^18^F-fluoro-β-D-glucose (18F-FDG), an analogue of glucose, with the concentrations of radiotracer accumulation in PET/CT image proportional to the metabolic activity of tissues concerning the underlying glucose accumulation and metabolism [166]. Fluorine 18–Sodium Fluoride (18F–NaF) is another radiotracer used more specifically for bone imaging in PET/CT, with the uptake proportional to blood flow in the bone and osseous remodeling [167,168,169]. Increased 18F-NaF bone uptake can be seen in an abnormal bone that undergoes higher remodeling, such as in osteoblastic or osteolytic processes and is used to differentiate various pathologies [170]. For the musculoskeletal system, the utility of PET/CT in distinguishing between malignant and benign bone tumours has also been widely studied and proved effective [171,172,173]. In fact, the metabolic information derived from PET/CT has been reported to provide a better characterization of bone lesions compared to conventional CT or MRI alone [29,174,175]. The standardized uptake value (SUV) technique with an optimal cut-off value of maximum SUV (SUVmax) is often the main feature used for the differential diagnosis of osseous tumours on a PET/CT [176]. However, the use of SUVmax alone as a distinguishing factor is often limited, due to the significant overlap of SUVmax values that occur between malignant vs. benign lesions [177,178,179,180]. Benign diseases, such as osteoarthritis, stress or trauma-related vertebral fractures, osseous hyperplasia, and underlying metabolic bone disease [181,182,183] have been reported to have high SUVmax values which may confound the diagnosis of malignant bone lesions.

To improve the current diagnostic efficacy of PET/CT interpretation, Fan et al. [184] utilized texture analysis with SUVmax to construct radiomics models to distinguish between benign vs. malignant bone lesions. Texture analysis extracts information, regarding the relationship between adjacent voxels or pixels, and assesses for inhomogeneity, which can then be used to predict the likelihood of benign vs. malignant bone lesions. [185,186]. By incorporating partial texture features along with SUVmax, the developed classification model (using logistic regression) achieved an accuracy of 87.5% compared to 84.3% for nuclear medicine physicians (*p* = 0.03) in differentiating spinal osseous metastases from benign osseous lesions with high SUVmax values.

In another study using 18F-NaF PET/CT imaging, Perk T et al. [87] created a machine tool for the automated classification of benign vs. malignant osseous lesions in patients with metastatic prostate cancer who received whole-body 18F-NaF PET/CT scans. The group analyzed up to 1751 bone lesions from a total of 37 subjects. The model included an analysis of 172 imaging and spatial probability features which showed superior classification performance compared to using SUVmax alone, with an AUC of up to 0.95. In addition, the model was also able to replicate the nuclear medicine physicians’ classification of bone lesions which had an AUC range of 0.91–0.93. This machine learning tool may potentially assist physicians in swiftly and accurately detecting and classifying bone lesions in 18F-NaF PET/CT scans.

### 4.5. Potential Clinical Impact and Applications

There is significant clinical value in the ability of machine learning to differentiate between benign vs. malignant bone lesions. A retrospective study by Stacy et al. [187] found that at least a third of patients with bone lesions referred to orthopedic oncology in a year had images that were diagnosed by radiologists as characteristic of benign tumours or non-malignant entities that did not require follow-up or referrals. Accurate AI models for the characterization of osseous lesions could therefore potentially reduce the rate of unnecessary specialist referrals and follow-up, reducing the associated healthcare costs and patient anxiety regarding a possible cancer diagnosis.

Moreover, more accurate characterizations of bone lesions would be valuable for radiologists to identify high-risk bone lesions that will benefit from biopsy to rule in malignancy with greater certainty. Unnecessary biopsies of benign bone lesions leave patients at risk of post-procedural complications, and hasty biopsy planning can increase the risk of misdiagnosis, creating unwarranted patient stress [188,189]. Biopsies can also be non-diagnostic in up to 30% of bone lesions, which requires repeat biopsies and an associated higher risk of complications [190]. A robust AI model that can identify benign bone lesions with a high specificity could help reduce the rate of unnecessary biopsies. This would be especially helpful for bone tumour subtypes which demonstrate similar imaging and histology features for their benign and malignant counterparts, for example, cartilaginous tumours including enchondroma (benign) vs. chondrosarcoma [135,165]. Recently, a CT-based radiomics model derived by Gitto et al. [78] was able to distinguish between ACT from high-grade chondrosarcoma with accuracy higher than that of pre-operative biopsy (81%, AUC 0.89 vs. 69%, AUC 0.66) albeit not reaching statistical significance (*p* = 0.29). Cartilaginous tumour characterisation remains a clinical challenge as wide resection is often required for definitive diagnosis and biopsy may inadvertently lead to tumour down-grading in lesions with marked heterogeneity, as only small areas can be sampled [166]. In addition biopsy of high-grade chondrosarcoma may inadvertently result in spillage or biopsy tract contamination [167,168].

On top of differentiating bone lesions into benign vs. malignant, machine learning methods can also help differentiate lesions secondary to post-treatment change from residual or recurrent malignant disease. This is a crucial diagnostic challenge as the treatment for the two entities are vastly different, and accurate diagnosis is also important to prevent unnecessary invasive biopsy and/or chemoradiotherapy. However, radiological evaluation of bone tumours after treatment can be quite difficult [191,192]. Zhong et al. [73] created an MRI-derived radiomics nomogram that was found to be clinically useful in discriminating between cervical spine osteoradionecrosis following radiotherapy and metastases with an AUC of 0.72 in the validation set. Another study by Acar E. et al. [78] utilized machine learning techniques via texture analysis on 68Ga-prostate-specific membrane antigen (PSMA) PET/CT images. These techniques were able to distinguish between sclerotic bone lesions with complete post-treatment response vs. metastatic bone disease with an AUC of 0.76.

The use of multiple imaging modalities to evaluate bone tumours is known to improve the accuracy of diagnosis [3]. Machine learning models built using different modalities (multimodal) have also been shown to improve diagnostic performance [193]. In breast radiology, Antropova et al. [194] came out with a CNN method involving fusion-based classification using dynamic contrast enhanced-MRI, full-field digital mammography, and ultrasound. This method outperformed conventional CNN-based and CADx-based classifiers, with an AUC of 0.90. In prostate cancer imaging, Sedghi A. et al. [195] developed models of fully convolutional networks integrating data from both temporal enhanced ultrasound along with the apparent diffusion coefficient (ADC) from MRI studies. The multimodality integration of information from both MRI and ultrasound achieved an AUC of 0.76 to 0.89 for the detection of prostate cancer foci, which outperformed the average unimodal predictions (AUC of 0.66–0.70). Future studies for bone tumours could adopt similar multimodal methods to further improve diagnostic accuracy, for example, CNN models using matched CT and MRI data to predict malignant potential.

## 5. Conclusions

Machine learning techniques to discriminate bone lesions have achieved a relatively good performance across various imaging modalities, demonstrating high sensitivity, specificity, and accuracy for distinguishing between benign vs. malignant lesions in several cohort studies. These techniques could improve the management of bone tumours in two key ways. Firstly, benign lesions could be targeted for an imaging follow-up rather than a biopsy, which will reduce unnecessary referrals to specialist clinics and prevents biopsy complications. Secondly, suspected malignant lesions could be targeted for expedited orthopaedic oncology referral, and additional machine learning tools could aid in determining the tumour subtype and location for the highest biopsy yield. Currently, the majority of studies are in the preliminary stage, limited by small sample sizes and the retrospective nature of the analyses. Further research is required, especially through the use of multicenter external datasets to assess the generalisability of the machine learning techniques on a greater range of imaging data. Multimodal machine learning techniques are an especially exciting area of future work for bone tumour characterization as they can fuse the information from different complementary modalities (for example, radiographs, CT, MRI and PET/CT) with clinical data, which is similar to the current radiologist assessment and could yield even greater accuracy.

## Figures and Tables

**Figure 1 cancers-15-01837-f001:**
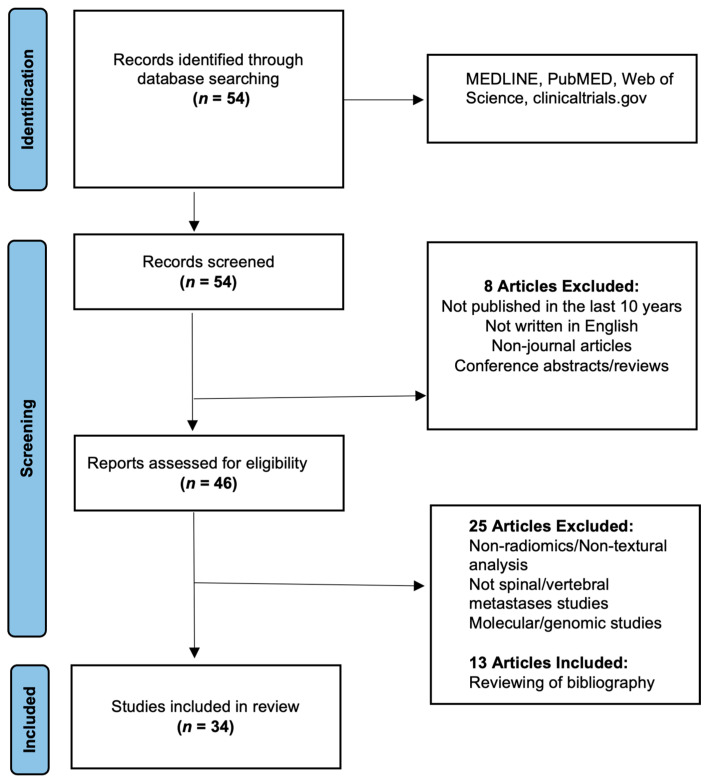
Preferred Reporting Items for Systematic Reviews and Meta-Analyses (PRISMA) flowchart for the study (adapted from the PRISMA group, 2020), which describes the research article selection process.

**Figure 2 cancers-15-01837-f002:**
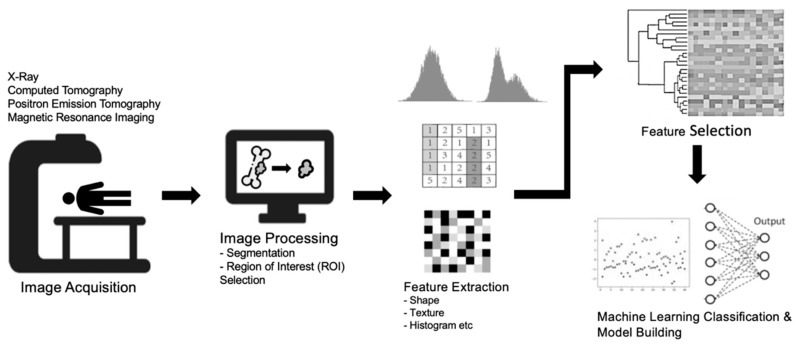
A diagram highlighting the general framework and key steps for radiomics, namely image acquisition, image processing (including segmentation), feature extraction in the curated regions of interest (ROIs), feature selection, exploratory analysis, and finally modelling.

**Figure 3 cancers-15-01837-f003:**
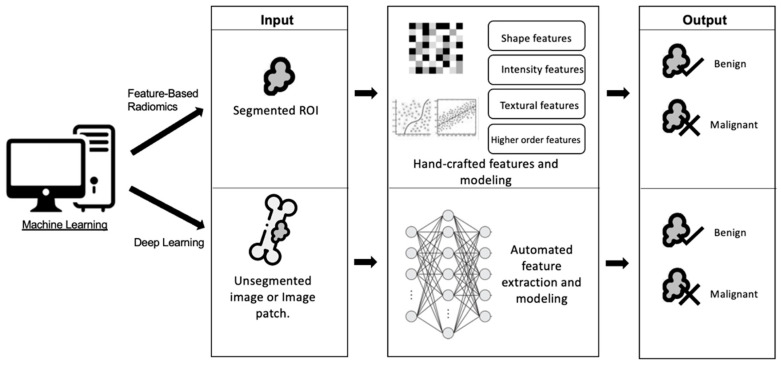
A diagram illustrating the difference between the two most popular techniques for machine learning; feature-based radiomics vs. deep learning for classifying lesions.

**Table 1 cancers-15-01837-t001:** Key characteristics of the selected research articles.

Authors	Artificial Intelligence Method	Publication Year	Main Objectives	Title of Journal	Main Imaging Modality Used	Performance
Xiong et al. [71]	Machine learning–based classifiers (ANN)	2021	Differentiate Multiple Myeloma from Metastases	Frontiers in Oncology	MRI	MCC = 0.605; Accuracy: 0.815; Sensitivity 0.879; Specificity: 0.790
Eweje et al. [72]	EfficientNet-B0 architecture and a logistic regression model (CNN)	2020	Differentiate between benign and malignant bone lesion	EBioMedicine	MRI	Accuracy: 0.760; Sensitivity: 0.790; Specificity: 0.750; AUC: 0.820
Zhong et al. [73]	Radiomics using MRI machine learning techniques (SVM)	2020	Differentiate between osteoradionecrosis from spinal metastases in Nasopharyngeal Carcinoma	BMC Medical Imaging	MRI	Accuracy: 0.737; Sensitivity: 0.843; Specificity 0.614; AUC of 0.725
Sun et al. [74]	Radiomics using CT machine learning techniques (SVM)	2021	Distinguish between benign and malignant bone tumours	Cancer Imaging	MRI	AUC of 0.892; NRI of 0.238; IDI of 0.163
Reinus et al. [75]	Two layer feed-forward neural network (ANN)	1994	Diagnosis of focal bone lesion	Investigative Radiology	X-ray	Accuracy 0.835
Filograna et al. [76]	Radiomics using MRI machine learning techniques (SVM)	2019	Differentiate between metastatic vs. non-metastatic vertebral bodies	La Radiologia Medica	MRI	AUC: 0.841–0.912
Yin P. et al. [77]	Radiomics using MRI machine learning techniques	2019	Differentiation of lesions in the sacrum, for example, chordoma, giant cell tumour, or metastatic lesions)	Journal of Magnetic Resonance Imaging	MRI	Accuracy: 0.810; AUC: 0.840
Acar E. et al. [78]	Texture analysis and PET CT machine learning techniques (Weighted KNN algorithm)	2019	Differentiating metastatic and completely responded sclerotic bone lesions in prostate cancer	British Journal of Radiology	PET/CT	Accuracy: 0.735; Sensitivity: 0.735; Specificity: 0.737; AUC: 0.76
He et al. [79]	EfficientNet-B0 convolutional neural network architecture	2020	Differentiating benign between benign and malignant bone lesion	EbioMedicine	X-ray	Accuracy: 0.734 AUC: 0.877 (benign); 0.916 (malignant) Accuracy: 0.99 Average mean IoU: 0.848
Reicher et al. [80]	TensorFlow Inception-v3 recurrent convolutional neural network (CNN) trained to the ImageNet model	2018	Classifying bone tumour matrix	SIIM (Society for Imaging Informatics in Medicine)	X-ray	Accuracy 0.93
Y. Li et al. [81]	Super label guided convolutional neural network	2018	Distinguish between benign and malignant bone tumours and classifies bone tumour matrix	Artificial Neural Networks and Machine Learning	CT	Accuracy: 0.740
Park et al. [82]	ResNet 50, GoogleNet Inception v3, and EfficientNet-b1, b2, and b3	2022	Distinguish between benign, malignant or no tumour over the femur	PLoS ONE	X-ray	Accuracy: 0.853; Sensitivity: 0.822; Specificity: 0.912; Precision: 0.821; AUC: 0.953
Liu et al. [83]	PyTorch 1.2.0 with Python 3.7.3, XGBoost and SHapley ad- ditive exPlanations value	2021	Classification of benign, intermediate and malignant bone lesions	European Radiology	X-ray	AUC: 0.898 (benign); 0.894 (malignant); 0.865 (intermediate) Macroaverage AUC: 0.872
Pan D. et al. [84]	Radiomics using X-ray machine learning techniques, SHapley Additive exPlanations (SHAP)	2021	Classification of benign, intermediate and malignant bone lesions	Biomed Research International	X-ray	AUC: 0.970 (binary), 0.940 (tertiary); Accuracy: 0.947 (binary); 0.828 (tertiary)
Hong J. H. et al. [85]	Radiomics using CT machine learning techniques	2021	Distinguish between benign bone island and osteoblastic metastasis	Radiology	CT	AUC: 0.960; Sensitivity: 0.800; Specificity: 0.960; Accuracy: 0.860
von Schacky et al. [86]	Radiomics using X-ray machine learning techniques, Python 3.7.7, scikit- learn 0.22.2 andfastai library	2022	Distinguish between benign and malignant bone tumours	European Radiology	X-ray	Accuracy: 0.80 (internal), 0.75 (external); Sensitivity: 0.75, 0.90; AUC: 0.79, 0.90
T. Perk et al. [87]	Radiomics using PET/CT machine learning techniques	2018	Distinguish between benign and malignant bone tumours	Physics in Medicine and Biology	PET/CT	AUC: 0.950; Sensitivity: 0.880; Specificity: 0.890
Bao H. D. et al. [88]	Radiomics using X-ray machine learning techniques	2017	Classification of different bone tumours	Journal of Digital Imaging	X-ray	Primary accuracy: 0.440–0.620; Differential accuracy: 0.620–0.800
Kahn et al. [89]	Bayesian network using X-ray machine learning techniques	2001	Classification of different bone tumours	Journal of Digital Imaging	X-ray	Accuracy: 0.890 (binary), 0.680 (tertiary)
Gitto et al. [90]	Radiomics using MRI machine learning techniques	2022	Distinguish between benign vs. malignant cartilaginous lesions	EBioMedicine	MRI	Accuracy: 0.98 (ACT), 0.80 (CS2); AUC: 0.94 (ACT), 0.90 (CS2)
Gitto et al. [91]	Radiomics using MRI machine learning techniques	2020	Distinguish between low-grade vs. high-grade cartilaginous lesions	European Journal of Radiology	MRI	Accuracy: 0.857 (training), 0.750 (test); AUC: 0.850 (training), 0.78 (test)
Yin et al. [92]	Radiomics using CT machine learning techniques, Pyradiomics python package,	2020	Distinguish between benign vs. malignant sacral tumour	Frontier Oncology	CT	Accuracy: 0.81 (Clinical-LR), 0.81 (Clinical-DNN); AUC: 0.84 (Clinical-LR), 0.83 (Clinical-DNN)
Georgeanu et al. [93]	Radiomics using MRI machine learning techniques, ResNet50 image classifiers	2022	Distinguish between benign and malignant bone tumours	Medicina (Kaunas)	MRI	Accuracy: 0.808 (training), 0.805 (test); AUC: 0.885 (training), 0.879 (test)
Fan et al. [34]	Radiomics using PET/CT machine learning techniques	2021	Distinguish between benign and metastatic vertebral lesions	Frontiers in Medicine	PET/CT	Accuracy: 0.875 (LR), 0.834 (SVM), 0.750 (Decision Tree)
Xu et al. [94]	Radiomics using PET/CT machine learning techniques	2014	Distinguish between malignant and benign bone and soft-tissue lesions	Annals of Nuclear Medicine	PET/CT	Accuracy 0.825; Sensitivity 0.864; Specificity 0.772
Chianca et al. [95]	Radiomics using MRI machine learning techniques, 3D Slicer heterogeneityCAD module (hCAD) and PyRadiomics	2021	Distinguish between different benign, primary malignant vs. metastatic vertebral lesions	European Journal of Radiology	MRI	2-Label Classification—Accuracy: 0.94 (training), 0.86 (test); 3-Label Classification—Accuracy: 0.80 (training), 0.69 (test).
Gitto et al. [96]	Radiomics using MRI machine learning techniques	2022	Distinguish between benign and metastatic vertebral lesions	La Radiologica Medica	MRI	Accuracy—0.76; Sensitivity: 0.78; Specificity: 0.68; AUC 0.78
Consalvo et al. [97]	PyTorch 1.9.0 and cuda toolkit 11.1 & ResNet18 architecture	2022	Distinguish between Ewing Sarcoma (ES) vs. acute osteomyelitis (OM)	Anticancer Research	X-ray	Accuracy: 0.867 (ES), 0.903 (OM); Sensitivity: 1.00 (ES), 0.930 (OM); Specificity: 0.76 (ES), 0.844
Zhao et al. [98]	Radiomics using MRI machine learning techniques	2022	Distinguish between benign vs. malignant bone lesions	Journal of Magnetic Resonance Imaging	MRI	Improved Sensitivities 0.12 to 0.36 as compared to manual.
Bradshaw et al. [99]	Deep convolutional neural network via VGG19 architecture	2018	Classifying benign and malignant bone lesion	Journal of Nuclear Medicine	PET/CT	Accuracy: 0.88; Sensitivity: 0.90; Specificity 0.85.
Do et al. [100]	Deep convolutional neural network with combination of global and patch-based models	2021	Classifying bone tumours in the knee into benign vs. malignant	Diagnostics	X-ray	Accuracy: 0.99 Average mean IoU: 0.848
Masoudi et al. [101]	Deep convolutional neural network with 2D ResNet- 50 & 3D ResNet-18	2021	Classify benign or malignant bone lesions in prostate cancer	IEEE Access	CT	Accuracy: 0.922; F1: 92.3%
Gitto et al. [102]	Machine-learning classifier (LogitBoost)	2021	Classification of atypical cartilaginous tumours and higher-grade chondrosarcoma, of long bones.	EBioMedicine	CT	Accuracy 0.750, AUC 0.78 (Validation set)
von Schacky et al. [103]	Mask region–based convolutional neural network (Mask-RCNN-X101)	2021	Classify benign or malignant bone lesions	Radiology	X-ray	Accuracy: 80.2%, Sensitivity: 62.9%, Specificity: 88.2%

Abbreviations: ANN (Artificial Neural Network); MCC (Matthews correlation coefficient); CNN (Convolutional Neural Network); AUC (Area under curve); SVM (Support vector machine); NRI (Net reclassification index; IDI (Integrated discrimination index); IoU (Index of Uniformity); DNN (Dense neural network) LR (Likelihood ratio).

**Table 2 cancers-15-01837-t002:** The performance of Artificial intelligence among various imaging modalities in distinguishing benign vs. malignant bone lesions.

Imaging Modality	Accuracy	Sensitivity	Specificity	Area under Curve (AUC)
X-ray	0.44–0.99	0.75–1.00	0.78–0.91	0.79–0.95
Computed Tomography (CT)	0.74–0.92	0.80	0.96	0.78–0.96
Magnetic Resonance Imaging (MRI)	0.74–0.98	0.78–0.88	0.61–0.79	0.73–0.94
Positron Emission Tomography with CT (PET/CT)	0.74–0.88	0.84–0.90	0.74–0.85	0.76–0.95
Overall	0.44–0.99	0.63–1.00	0.73–0.96	0.73–0.96

## Data Availability

Not applicable.

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
