# Peer review of "Application of Machine Learning for Differentiating Bone Malignancy on Imaging: A Systematic Review"

_cancers, 2023, doi:10.3390/cancers15061837_

Round 1

Reviewer 1 Report

The paper is well written and structured. Besides, the origination structure of this paper is complete. I think this paper is acceptable after some minor revisions.

1. It would be better to make the contributions of this work more clear. In the main body of this paper, the presentation of the idea advocated by the authors is mixed with the existing works. It should be better that the authors to point out their contributions more clearly (especially, the theoretical contributions).

2. The rule for acronyms is to first spell out the reference, followed by the acronym in parentheses, at its first use. Refer to the acronym thereafter, but repeat the full spelling and acronym at its first use in any major section. You have not yet defined this one.

3. More clarifications and highlights about the research gaps in the related works section needs to be included.

Author Response

Response to Reviewer 1:

Dear Editors and reviewers,

Thank you for taking the time to review our manuscript. We have made the necessary point by point responses to the reviewer comments and made the necessary edits to the manuscript.

Reviewer 1:

R1.1. It would be better to make the contributions of this work more clear. In the main body of this paper, the presentation of the idea advocated by the authors is mixed with the existing works. It should be better that the authors to point out their contributions more clearly (especially, the theoretical contributions).

Response: Thank you for the comments. We have edited the Author’s contribution segment to further elaborate on each doctor’s contribution to the paper. We have also added an additional segment on the discussion points.

R1.2. The rule for acronyms is to first spell out the reference, followed by the acronym in parentheses, at its first use. Refer to the acronym thereafter, but repeat the full spelling and acronym at its first use in any major section. You have not yet defined this one.

Response: Thank you for the suggestion. As per your recommendations, we have edited accordingly, spelling out the acronym in full at the start of each major section.

R1.3. More clarifications and highlights about the research gaps in the related works section needs to be included.

Response: Thank you for the comment. We have added clarifications and elaborated on the research gaps and limitations of the related works section under the discussion portion.

Thank you for your consideration of this manuscript.

Yours sincerely,

Wilson Ong

Department of Diagnostic Imaging, National University Health System, Singapore

Reviewer 2 Report

see the report

Author Response

Response to Reviewer 2:

Dear Editors and reviewers,

Thank you for taking the time to review our manuscript. We have made the necessary point by point responses to the reviewer comments and made the necessary edits to the manuscript.

Reviewer 2:

R2.1. in Introduction in lines 100-104 some applications for different tumors are presented. I suggest the inclusion of a brief description of eXplainable Artificial Intelligence for the aforementioned interpretability issue, as made for breast cancer in “Analyzing breast cancer invasive disease event classification through explainable artificial intelligence, Frontiers in medicine (2023)”, which could be helpful in the description of features interplays, as those introduced in lines 87-91.

Response: Thank you for the comments. We have added the recommended article in our references and included a segment on eXplainable artificial intelligence and its utility in oncology imaging (in lines 102-106). A brief description of features interplay, type of feature extraction and further elaboration on the AI models was described in Segment Header 3.2. We agree that adding a brief description of features interplays will be useful, although different studies used different methods and various feature extraction models which makes it difficult for us to further elaborate how each feature was extracted and selected accordingly. We have also summarised the broad steps into Figure 2 and 3.

R2.2. in lines 168-176 there is a statistical description of the performances associated with each imaging modality, that should be resumed in a table;

Response: Thank you for the suggestion. We have summarised the results in a new table 2.

R2.3. in the same lines and later on you describe some studies reaching a sensitivity equal to 1, which could be explained in terms of the corresponding dataset misbalance? The information about datasets benign and negative cases characterizing each study mentioned in the resuming table is missing and it should be included;

Response: Thank you for the comments. We note that within the conventional radiography portion, one of the studies achieved sensitivity = 1.00. This was because the study by Consalvo et al. which focused on mainly differentiating Ewing sarcoma from acute osteomyelitis used healthy and “normal” radiographs as a control group. We have provided an explanation in lines 186-190.

R2.4. in lines 176-178 the statement concerning multi-label classification problems should be supported by some metrics evaluation.

Response: Thank you for the comments. We have included details on the metrics evaluation in lines 197-203.

Thank you for your consideration of this manuscript.

Yours sincerely,

Wilson Ong

Department of Diagnostic Imaging, National University Health System, Singapore
